# Unique advantages of dynamic L-[11C] methionine PET/CT for assessing the rate of skeletal muscle protein synthesis: A pilot trial in young men

Koichiro Sumi[1‡], Kana Yamazaki[2‡*], Ryuichi Nishii[2,3], Misato Sakuda[1], Kentaro Nakamura[1], Kinya Ashida[1], Kentaro Tamura[2], Tatsuya Higashi[2]

**1** R&D Division, Meiji Co., Ltd., Hachiouji, Tokyo, Japan, **2** Department of Molecular Imaging and Theranostics, Institute for Quantum Medical Science, National Institutes for Quantum Science and Technology, Inage, Chiba, Japan, **3** Biomedical Imaging Sciences, Department of Integrated Health Sciences, Nagoya University Graduate School of Medicine, Tokai National Education and Research System, Higashi-ku, Nagoya, Japan

‡ KS and KY contributed equally to this work and are first authorship on this work.
* yamazaki.kana@qst.go.jp

**Data Availability Statement:** The anonymized data that support the findings of this study are available

## Abstract

Although the standard method to evaluate skeletal muscle protein synthesis (MPS) is muscle biopsy, the method is invasive and problematic for multisite use. We conducted a small pilot study in volunteers to investigate changes in MPS according to skeletal muscle site using a noninvasive method in which 6 healthy young men were given yogurt (containing 20 g milk protein) or water, and 1 h later, L-[11C]methionine ([11C]Met) was administered intravenously. Dynamic PET/CT imaging of their thighs was performed for 60 min. The influx constant $K_i$ of [11C]Met in skeletal muscle protein was calculated as an index of MPS using a Patlak plot, and found to be 0.6%–28% higher after ingesting yogurt than after water in 5 of the 6 volunteer participants, but it was 34% lower in the remaining participant. Overall, this indicated no significant increase in $K_i$ after ingesting milk protein. However, when the quadriceps and hamstring muscles were analyzed separately, we found a significant difference in $K_i$. This demonstrates the potential of visualizing MPS by calculating the $K_i$ for each voxel and reconstructing it as an image, which presents unique advantages of [11C]Met PET/CT for evaluating MPS, such as site-specificity and visualization.

## Introduction

Skeletal muscles play an important role not only in improving the performance of athletes but also in maintaining health in the general population [1–5]. Skeletal muscle protein synthesis (MPS) is essential for the growth and homeostasis of skeletal muscle [6]. In recent years, age-related muscle atrophy, known as sarcopenia, has become a major issue in our aging society. Therefore, maintaining healthy skeletal muscles could improve overall well-being.

in the Zenodo public repository with the Digital Object Identifier (DOI) "10.5281/zenodo.10993747.

**Funding:** The author(s) received no specific funding for this work.

**Competing interests:** I have read the journal's policy and the authors of this manuscript have the following competing interests: [K. S., M. S., K. N., and K. A. are salaried employees of Meiji Co., Ltd., a food company, which also provided experimental funding, materials, facility, and staff assistance to this research. But, Meiji Co., Ltd. did not play any role in the study design, data collection and analysis, decision to publish, or preparation of the manuscript. K. Y., R. N., K. T., and T. H. has no conflicts of interest to declare.]

Quantification of the impact of lifestyle changes and medical interventions on MPS in sarcopenia is a major focus of skeletal muscle research.

Currently, the criterion standard for MPS measurement involves directly quantifying the incorporation of stable isotope-labeled amino acids into skeletal muscle proteins [7]. While this method is highly reliable, it is not easy to apply in frail individuals, such as older adults and patients under long-term care, because repetitive muscle biopsy is required. To address this limitation, we focused on dynamic positron emission tomography/computed tomography (PET/CT) using L-[$^{11}$C]methionine ([$^{11}$C]Met) as a tracer to assess MPS. The use of [$^{11}$C]Met PET/CT is a minimally invasive, clinically-applicable method of evaluating MPS and has been developed through animal experiments [8, 9] and clinical studies [10, 11] since the late 1990s. More recently, Arentson-Lantz et al. demonstrated that MPS upregulation after whey protein ingestion in the elderly can be assessed using [$^{11}$C]Met PET/CT, similarly to the conventional method of MPS measurement using biopsy-based L-[ring-$^{13}$C$_6$]-phenylalanine kinetics [12].

Due to the nature of muscle biopsy, the clinical target for evaluating MPS is limited mainly to large surface muscles such as the quadriceps femoris [12]. It is especially difficult to evaluate the hamstrings, which are in a deeper area, using biopsy. By contrast, PET/CT enables MPS to be evaluated in the whole thigh region, including in the deeper layers [10–12]. Preclinical animal experiments have shown that not only the basal MPS but also increased MPS following stimuli such as exercise or nutrition varies depending on the properties of the skeletal muscles, such as the proportion of slow-twitch or fast-twitch muscles [13–16]. Therefore, for clinical application, the ability to detect differences in MPS and analyze specific muscle regions would be additional advantages of evaluating MPS using PET/CT [17].

Further to the study by Arentson-Lantz et al. [12], we used [$^{11}$C]Met PET/CT in this pilot study to investigate changes in MPS in the entire middle-thigh region of young men, following their ingestion of milk protein, which is generally composed of 20% whey and 80% casein. Furthermore, we analyzed the quadriceps and hamstrings separately to determine whether there was a difference in MPS between these muscle groups.

## Materials and methods

### Approval and registration

The protocol of this study was conducted in compliance with the 1964 Declaration of Helsinki. This clinical study was reviewed and approved by the Institutional Review Board of National Institutes for Quantum and Radiological Science and Technology (QST) and Meiji Co. (approval No. #19–004), and it was registered in the University Hospital Medical Information Network Clinical Trials Registry: UMIN-CTR data base (UMIN000037199). Written informed consent was obtained from each participant.

### Participants

Healthy young men aged 20–40 years with a body mass index (BMI) of 18.5–25.0 kg/m$^2$ were recruited by a public clinical research volunteer recruitment system administered by QST from June 28, 2019 to November 29, 2019. Volunteers who met the following criteria were excluded: (1) receiving medication on the test day, (2) severe or progressive diseases such as malignancies, (3) obvious abnormalities on physical or laboratory examinations, (4) milk allergy or lactose intolerance, (5) smokers, (6) engaging in habitual moderate to high intensity exercise, (7) claustrophobia, (8) participation in other clinical trials within the previous month, and (9) prior examination involving radiation exposure for medical treatment purposes within the past 12 months or for research purposes within the past 6 months.

## Procedures

This pilot study aimed to incorporate 6 analyzable participants as a minimum sample size for statistical analysis, and was designed as a two-group, two-period, open-label crossover study because we estimated that there are large individual differences in basic MPS and MPS responsiveness to stimulation. On the day of the first test, after an overnight fast, the candidates arrived 2–3 h prior to the PET/CT evaluation and underwent screening tests (blood analysis, medical interviews, and a questionnaire on lifestyle habits). The volunteers who passed the screening tests were included in the study. The clinical trial coordinator, who was not directly involved in research, randomly allocated the volunteer participants to one of two groups using sealed envelopes. The allocation was concealed until interventions were assigned. The participants then ingested the test food, which was 200 mL water (fasting state) or 200 g commercial low-fat drained (known as Greek-style) yogurt (containing 20.4 g milk protein, 0.3 g fat, and 9.5 g carbohydrates; Meiji Co., Tokyo, Japan), within 10 min. At 1 h after ingestion of the test food, PET/CT was performed according to the method described below. The second test was performed 6–14 days after the first test. The half-life of [$^{11}$C]Met is approximately 20 min, so a washout period of 6 days or more is sufficient. To avoid changes in the condition of the participants, including lifestyle changes, the interval between the first and second tests was within two weeks. The study protocol of the second test was the same as that of the first test, except that there was no screening test, and an additional test food not included in the first test was ingested. All tests and data collection were conducted at QST hospital. An overview of the test scheme is shown in Fig 1A.

**PET/CT image acquisition.**   PET/CT images were acquired using the Aquiduo PCA-700B (Canon Medical Systems, Tochigi, Japan) or Biograph 16 (Siemens Healthcare, Nashville, TN, USA) scanner. The same PET/CT scanner was used for both tests on the same participant.

[$^{11}$C]Met was synthesized in our laboratory on the day of the test. Radiochemical purity of the [$^{11}$C]Met exceeded 99% and molecular activity was 162.2 ± 77.6 GBq/μmol. The participants were kept in a resting state for 1 h after ingestion of the test food, and after supplying 300 mL water for hydration, the participants were encouraged to urinate. Dynamic PET/CT using [$^{11}$C]Met to estimate the rate of protein synthesis based on the time activity curve analysis of the radiotracer was planned according to a method described previously by Arentson-Lantz et al. [12]. Participants entered the scanner in the supine position with their legs fixed. According to a previous study [12], 6.5–7.9 MBq/kg [$^{11}$C]Met in saline solution (484.1 ± 46.2 MBq) was administered intravenously as a bolus injection. Concurrently with the bolus administration of [$^{11}$C]Met, dynamic PET imaging of the femur was acquired for 60 min with a frame duration of 10 s × 15, 30 s × 5, 5 min × 4, 10 min × 2, and 15 min × 1.

Dynamic PET images of a total of 81 slices of the mid-thigh were collected every 2 mm over a range of approximately 16 cm. After dynamic PET imaging, whole-body CT for correction of PET attenuation was performed. Finally, whole-body static PET imaging was performed for 30 min at 60–90 min after administration of [$^{11}$C]Met to confirm the distribution of [$^{11}$C]Met in the body.

**PET/CT image analysis.**   We used an image-derived MPS estimation method to take full advantage of the non-invasive features of PET/CT. Corrections for decay with the physiological half-life of $^{11}$C (20.4 min), partial volume effect (PVE), and metabolites of methionine were applied (see following). Volumes of interest (VOIs) in the femoral artery (input region) and skeletal muscle tissue (target region) were drawn on the PET image. The time activity curve (TAC) of [$^{11}$C]Met in each VOI was extracted. The extracted input TAC was corrected for the resolution of the PET device and for the in vivo metabolites of [$^{11}$C]Met. Patlak graphical analysis was employed to estimate the influx constant $K_i$ using the corrected input function in the

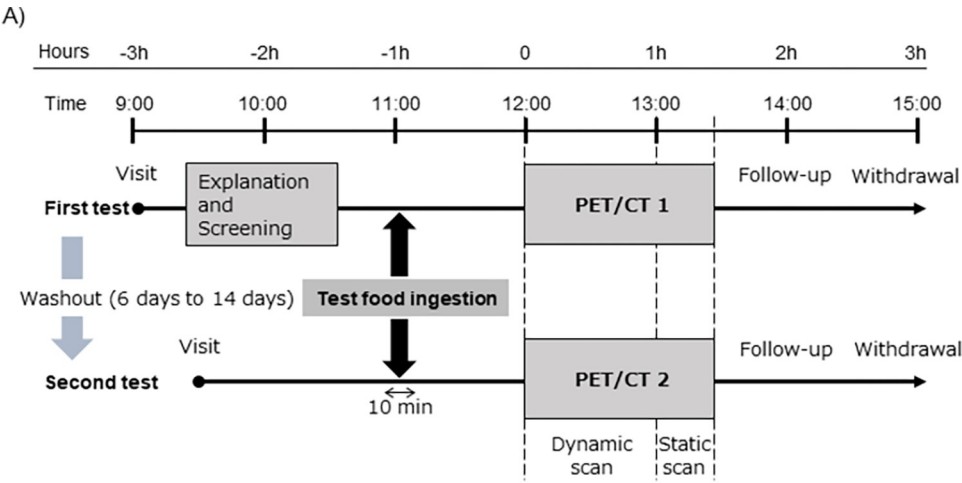

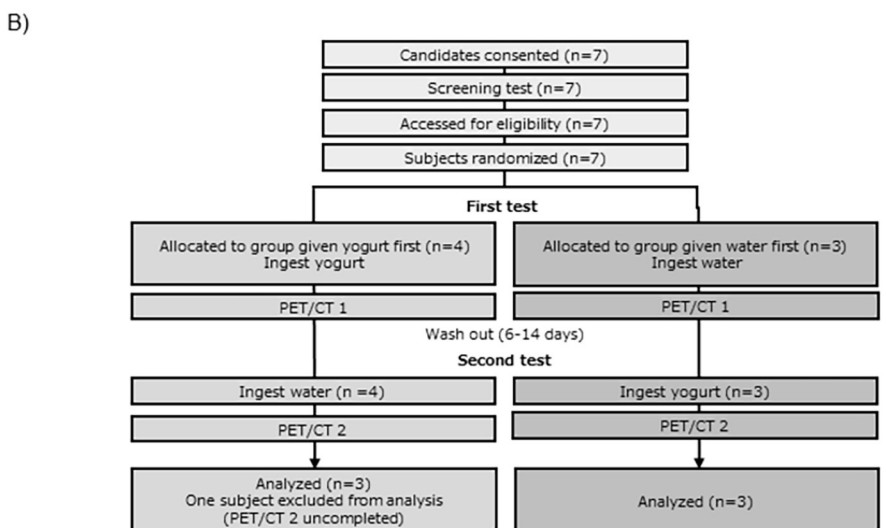

**Fig 1. Study procedure and participant flow.** (A) The participants ingested the test food (water or yogurt containing 20.4 g milk protein) and underwent PET/CT imaging. The second test was conducted 6–14 days after the first test, as a washout period. In the second test, an additional test food not included in the first test was ingested. (B) A flow diagram showing participants in this study. Seven healthy young men were registered and randomized into two groups, but one allocated in the group that was given yogurt could not complete the PET/CT 2 due to an urge to urinate. Therefore, the data from the 6 remaining participants were included in the analysis.

femoral artery and the concentration in the target region [18]. These methods were based on previous reports that image-derived $K_i$ without blood or tissue sampling can reflect increases in MPS due to clinically known anabolic stimuli (i.e., exercise [11] and protein ingestion [12]). An overview of the PET/CT image analysis is shown in Fig 2.

**VOI setting.** All VOIs were determined over an 8–9 cm length in the axial direction (40–45 slices) at the center of the 16-cm axial-length PET image (Fig 2A). The VOI of the femoral artery was defined according to Harnish et al. [11]. In brief, using the images collected when the radiation counts in the input part of the dynamic PET image were maximum (usually 30–50 s after tracer injection), the region with a radiation intensity ranging from 50% maximum to maximum was extracted from each transaxial slice (Fig 2B); by merging them, the cylindrical VOI was obtained. The VOI of skeletal muscle tissue was determined on fused PET/CT

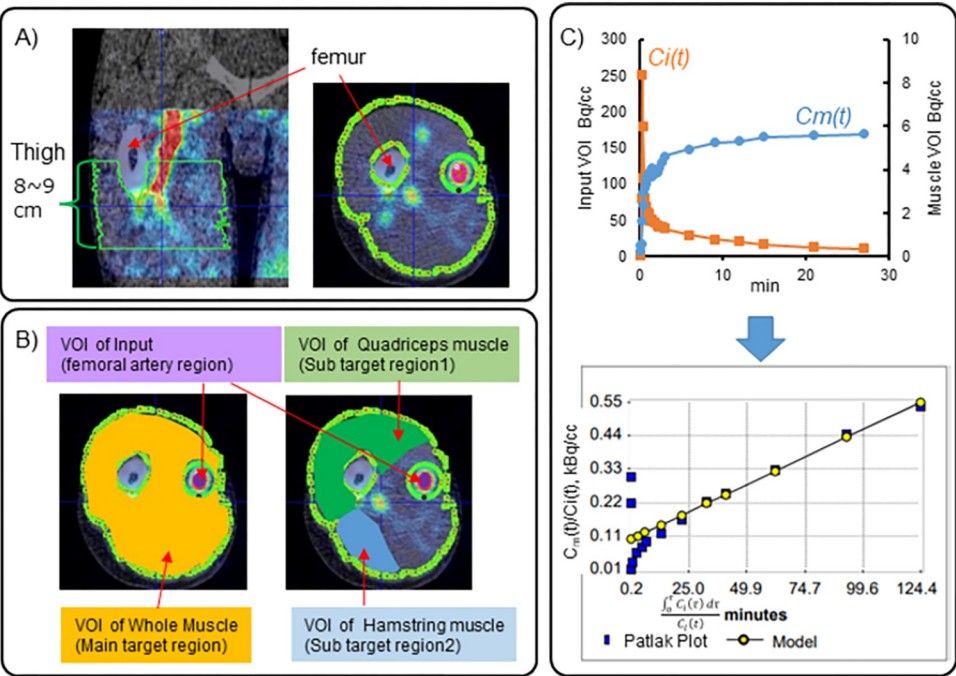

**Fig 2. Dynamic PET/CT analysis.** (A) VOI setting in the dynamic PET image of the mid-thigh. A coronal section (left) and axial section (right) are shown. (B) VOI setting in the whole or separated skeletal muscle and femoral artery on an axial section. (C) Calculation of $K_i$ derived from the slope of the Patlak plot. The skeletal muscle concentration ($C_m(t)$) and corrected femoral artery input function ($C_i(t)$) are shown at the top. A Patlak plot of $C_m(t)$ and $C_i(t)$ and an automated fitting model line using PMOD software is shown at the bottom.

images. That is, regions excluding subcutaneous adipose tissue and bone were excluded with reference to the CT image intensity. To eliminate the influence of radiation signal spill-in from the surrounding background into the femoral artery, a circular area with a diameter of 15 mm centered on the femoral artery was further excluded from the extracted area as the background VOI (Fig 2B). A VOI in whole muscle tissue was obtained by merging regions created in the same manner in each slice. VOIs in the quadriceps femoris and hamstrings within the whole femoral artery VOI were drawn manually by referring to the CT image, according to Bucci et al. [17] (Fig 2B). The PFUS module of PMOD software (ver. 3.9, PMOD Technologies, Fällanden, Switzerland) was used to draw the VOIs.

**TAC extraction and correction.** TAC data from each VOI were quantified using the PMOD software. Correction for PVE was performed to obtain a suitable image-derived input function as a substitute for repeated sampling of arterial blood [19]. Also, ¹¹C in arteries includes not only free [¹¹C]Met, which can be used for protein synthesis, but also its metabolites (e.g., albumin including [¹¹C]Met is rapidly produced and secreted from the liver) [20]. Therefore, metabolite correction was performed to estimate the input function originating from free [¹¹C]Met within the VOI of the femoral artery [8]. To ensure noninvasiveness, the correction constants estimated from previous reports were incorporated into the equation below. The input function $Cp(t)$, corrected for PVE, was calculated as:

$$C_p(t) = \frac{(C_{ART}(t) - a \times C_{BG}(t))}{b} \qquad \text{Eq [1]}$$

Where, $C_{ART}(t)$ is the input function; $C_{BG}(t)$ is the concentration obtained from the background VOI; and $a$ and $b$ are the spill-in and recovery coefficient constants to correct the

errors caused by the spatial resolution of the PET image. The spill-in coefficient constant $a$ is a function of the diameter $x$ of the femoral artery VOI. The recovery coefficient was fixed at $b = 0.1$, which was estimated from previously described data [19].

The spill-in coefficient $a$ is expressed as:

$$a = ((1 - c) \times \exp(-dx) + c) \qquad \text{Eq [2]}$$

Here, $c$ and $d$ are constants to reduce the effect of spill-in also caused by the spatial resolution of the PET image. In this study, these constants were estimated as $c = 0.156$ and $d = 0.0614$, according to a previous study [19]. Subsequently, metabolite correction was performed using Eq [3] to calculate the corrected input function $C_i(t)$, which is assumed to be derived mostly from free [$^{11}$C]Met.

$$C_i(t) = C_p(t) \times \left( 1 - f \times \left( 1 - exp\left( -t \times log\left(\frac{2}{g}\right) \right) \right) \right) \qquad \text{Eq [3]}$$

Eq [3] is expressed as a function of time $t$, with constant $f = 890$ and constant $g = 28666$, based on the results of [$^{11}$C]Met administration studies in Japanese individuals [20].

The calculations for these corrections were performed using Excel 2016 (Microsoft, Redmond, WA, USA).

**Calculation of $K_i$.** The corrected femoral artery input function ($C_i(t)$) and the skeletal muscle concentration ($C_m(t)$) were input into the PMOD software (PKIN module) to calculate the $K_i$. The Patlak plot was drawn with $C_m(t)/C_i(t)$ as the vertical axis and $\int C_i(\tau)d\tau/C_i(t)$ as the horizontal axis (Fig 2C). Then, the maximum error was set to 10%, linear fitting was performed on the plot after the point of inflection, and the $K_i$, which is the slope, was calculated.

**Voxel-by-voxel $K_i$ heatmapping.** We calculated voxel-by-voxel $K_i$ and interpreted its heatmap, as a non-invasive parametric image representing the MPS rate, using the PXMOD module in PMOD software.

**Screening blood tests.** Using whole blood samples, blood cell counts (red blood cells, white blood cells, and platelets), hemoglobin and hematocrit concentrations, and blood laboratory test results (total protein, albumin, AST, ALT, blood sugar, HbA$_{1c}$, urea nitrogen, and C-reactive protein concentrations) were measured in the screening.

**Statistical analysis.** All values are expressed as means ± SD. Differences between test days or skeletal regions were assessed by paired $t$ test. Correlations between each variable were evaluated by Pearson's correlation analysis. $P < 0.05$ was considered statistically significant. Statistical tests were performed using JMP 15 software (SAS Institute, Cary, NC, USA).

## Results

### Participants

Seven healthy men were registered in this study, but one could not complete PET due to the urge to urinate. The data from the 6 remaining participants were included in the analysis. No adverse events were observed in any participant. Participants were recruited from July 5 to November 22, 2019, and the intervention, including follow-up, was completed from July 5 to December 1, 2019. The participant flow, background characteristics and laboratory test results are shown in Fig 1B and Table 1, respectively.

### $K_i$ of [$^{11}$C]Met in the entire skeletal muscle of the thigh

$K_i$ of each individual in both thighs, the right thigh, and the left thigh after water or milk protein ingestion is summarized in Table 2. Fig 3 shows the mean $K_i$ on each side of the thigh

**Table 1.**

| Index | | |
|---|---|---|
| n | 6 | |
| Male, % | 6 (100%) | |
| Age | 24.0 | ± | 3.9 |
| Height, cm | 175.3 | ± | 7.5 |
| Weight, kg | 67.7 | ± | 9.0 |
| BMI, kg/m$^2$ | 22.0 | ± | 2.4 |
| Diastolic blood pressure, mmHg | 117 | ± | 10 |
| Systolic blood pressure, mmHg | 72 | ± | 6 |
| Blood biochemical analysis | | | |
| Total protein, g/dL | 7.2 | ± | 0.3 |
| Albumin, g/dL | 4.8 | ± | 0.3 |
| Urea nitrogen, mg/dL | 12.2 | ± | 4.0 |
| Creatinine, mg/dL | 0.86 | ± | 0.06 |
| Glucose, mg/dL | 96 | ± | 5 |
| AST, U/L | 22 | ± | 7 |
| ALT, U/L | 29 | ± | 16 |
| CRP, mg/dL | 0.02 | ± | 0.02 |
| HbA1c, % | 5.4 | ± | 0.3 |
| White blood cells, $10^2$/μL | 59 | ± | 17 |
| Red blood cells, $10^4$/μL | 510 | ± | 33 |
| Hemoglobin, g/dL | 15.0 | ± | 0.9 |
| Hematocrit, % | 43.2 | ± | 2.1 |
| Platelets, $10^4$/μL | 24.2 | ± | 4.6 |

after ingestion of water or milk protein. Individual $K_i$ and individual differences in $K_i$ between water and milk protein ingestion are also shown. There was no significant difference in the mean $K_i$ of the whole thigh muscle after ingesting water versus 20 g milk protein (Fig 3A). $K_i$ was 0.6–28% higher after ingesting milk protein than after ingesting water in 5 of the 6 participants. However, it was 34% lower in the remaining participant (Fig 3B). The correlation coefficient ($r$) between $K_i$ after water ingestion and that after milk protein ingestion was 0.178, which was not significant (Fig 3C). Fig 4 shows the $K_i$ calculated separately for the left and

**Table 2.**

| Participant (color of marker) | Sex | Age | Height, cm | Weight, kg | BMI, kg/m$^2$ | $K_i$ (both thighs) | | $K_i$ (right thigh) | | $K_i$ (left thigh) | | Note |
|---|---|---|---|---|---|---|---|---|---|---|---|---|
| | | | | | | Water | Milk protein | Water | Milk protein | Water | Milk protein | |
| A (red) | Male | 22 | 184.6 | 74.9 | 22.0 | 0.00503 | 0.00332 | 0.00543 | 0.00387 | 0.00478 | 0.00286 | Large difference in $Ki$ between the two test days |
| B (orange) | Male | 32 | 169.8 | 70.2 | 24.4 | 0.00365 | 0.00370 | 0.00355 | 0.00336 | 0.00376 | 0.00386 | |
| C (light blue) | Male | 22 | 182.2 | 80.3 | 24.2 | 0.00479 | 0.00501 | 0.00469 | 0.00543 | 0.00488 | 0.00462 | |
| D (yellow) | Male | 23 | 165.0 | 63.4 | 23.3 | 0.00354 | 0.00356 | 0.00341 | 0.00348 | 0.00339 | 0.00363 | |
| E (blue) | Male | 23 | 173.1 | 57.0 | 19.0 | 0.00306 | 0.00392 | 0.00364 | 0.00376 | 0.00240 | 0.00385 | Large difference in $Ki$ between the two test days |
| F (green) | Male | 22 | 176.9 | 60.5 | 19.3 | 0.00359 | 0.00411 | 0.00363 | 0.00386 | 0.00354 | 0.00386 | |

"Color of marker" is the color of the points for each corresponding participant in Figs 3 and 4.

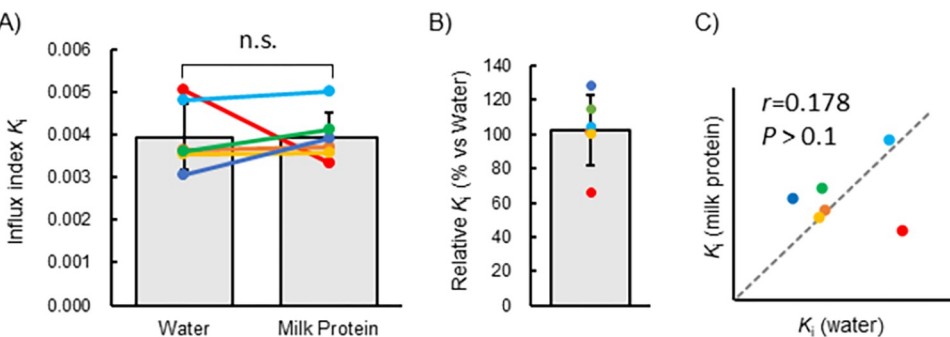

**Fig 3. Whole muscle $K_i$ after water or milk protein ingestion.** (A) $K_i$ of whole muscle regions. The bar graphs represent the means and standard deviations, with each colored dot indicating the value for each participant. No significant difference in the $K_i$ was observed between the test foods after ingestion. (B) $K_i$ after milk protein ingestion relative to $K_i$ after water ingestion. The bar graphs represent the mean and standard deviation, with each colored dot indicating the value for each participant. (C) Scatter plot representing $K_i$ after water ingestion and $K_i$ after milk protein ingestion. There was no significant correlation between the $K_i$ ($r = 0.178$, $P > 0.1$). The gray dotted line indicates where the $K_i$ were equal (slope is 1, i.e., equation expressed as y = x). Points of the same color indicate data from the same participant.

right thighs on each test day. In the first test, the difference in the left and right $K_i$ was $\leq 10\%$ in 2 of the 6 participants, $\leq 20\%$ in 3 participants, and 51% in 1 participant (Fig 4A). In the second test, 5 of 6 participants had a difference between the left and right thighs of $\leq 10\%$, whereas a 35% difference was observed in the remaining participant (Fig 4B). PET/CT images of the right thigh of participant A, whose input VOI position moved over time due to body motion, is shown in Fig 4C.

## $K_i$ of [$^{11}$C]Met in the quadriceps and hamstring muscles

Table 3 shows the $K_i$ of each individual for the quadriceps and hamstrings after water or milk protein ingestion. $K_i$ in the quadriceps and hamstrings and their correlation are summarized in Fig 5. $K_i$ was significantly higher in the quadriceps than hamstrings, both after water and milk protein ingestion (Fig 5A). Also, a high positive correlation of $K_i$ between the quadriceps and hamstrings was observed both after water ($r = 0.86$, $P < 0.05$) and milk protein ($r = 0.86$, $P < 0.05$) ingestion (Fig 5B).

### Visualization of protein synthesis by voxel-by-voxel $K_i$ heatmapping

As a noninvasive method, we calculated voxel-by-voxel $K_i$ and interpreted its heatmap, representing MPS. A three-dimensional voxel-by-voxel heatmap of $K_i$ fused with the CT image is shown in Fig 6. In this example, $K_i$ was relatively high in the quadriceps portion and relatively low in the hamstrings portion over the whole range, consistent with the above results.

## Discussion

In this pilot study, we successfully demonstrated a non-invasive imaging technique for estimating MPS. Additionally, we estimated and compared the differences in MPS after ingestion of milk protein and water. Notably, our study is the first to indicate that MPS may differ between the quadriceps and hamstrings by estimating site-specific $K_i$. Furthermore, we interpreted its heatmap image as a non-invasive parametric 3D image representing MPS rate. Our proposal is to acquire site-specific and/or non-targeted visualized MPS data from PET/CT images, not only to improve the condition of athletes, but also to devise countermeasures against sarcopenia in the elderly and individuals with chronic disease. There should be a

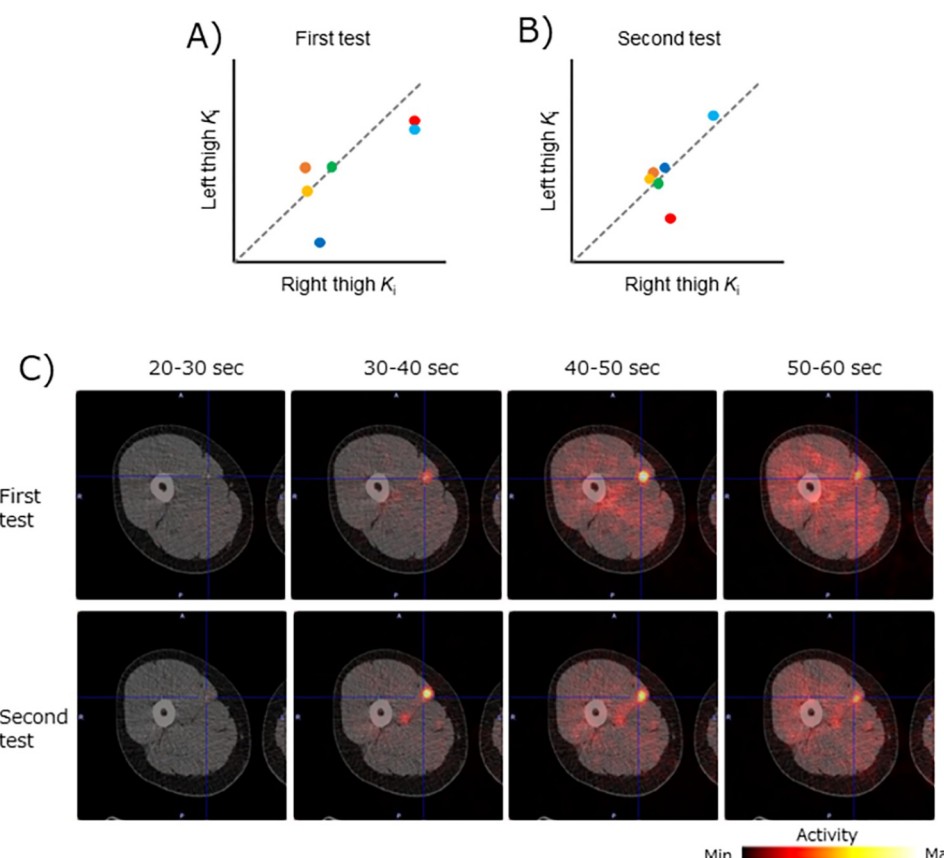

**Fig 4.** Left-to-right difference in $K_i$ in the first (A) and second (B) tests. The gray dotted lines in Fig 4A and 4B will overlap the marker if the left and right $Ki$ are equal. Participants A and E, whose relatively large body movements were observed during dynamic PET imaging, showed particularly large left-to-right differences. (C) Example of right-thigh axial section PET/CT of a participant with large body movements (Participant A). The intersection point of the blue vertical line and horizontal line in each image is presumed to be on the femoral artery.

sufficiently high demand for low-burden methods, such as PET/CT. The noninvasive evaluation of thigh muscles is an important advantage, because it prioritizes the comfort and safety of individuals by avoiding the need for biopsy.

Our first aim in this study was to non-invasively detect the deterministic physiological phenomenon such as the increase in MPS after ingestion of milk protein, using PET/CT. The

**Table 3.**

| Participant (color of marker) | $K_i$ (quadriceps) | | $K_i$ (hamstrings) | |
|---|---|---|---|---|
| | Water | Milk protein | Water | Milk protein |
| A (red) | 0.00550 | 0.00359 | 0.00427 | 0.00307 |
| B (orange) | 0.00429 | 0.00434 | 0.00283 | 0.00293 |
| C (light blue) | 0.00519 | 0.00543 | 0.00436 | 0.00447 |
| D (yellow) | 0.00391 | 0.00395 | 0.00320 | 0.00331 |
| E (blue) | 0.00319 | 0.00422 | 0.00296 | 0.00354 |
| F (green) | 0.00394 | 0.00452 | 0.00336 | 0.00367 |

"Color of marker" is the color of the points for each corresponding participant in Figs 3 and 4.

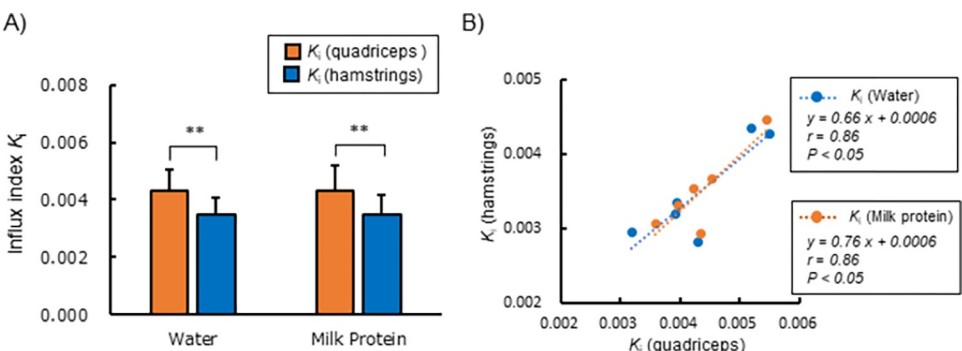

**Fig 5. Difference in each $K_i$ calculated between the quadriceps and hamstrings regions.** (A) Mean site-specific $K_i$ for the right and left thighs after water or milk protein ingestion. $K_i$ were significantly higher in the quadriceps than hamstrings after both milk protein and water ingestion (paired t-test, **P < 0.01). (B) Correlation between the $K_i$ of the quadriceps and that of the hamstrings. A positive correlation was observed between the two values after ingestion of both milk protein and water (Pearson's correlation coefficients and P values are shown).

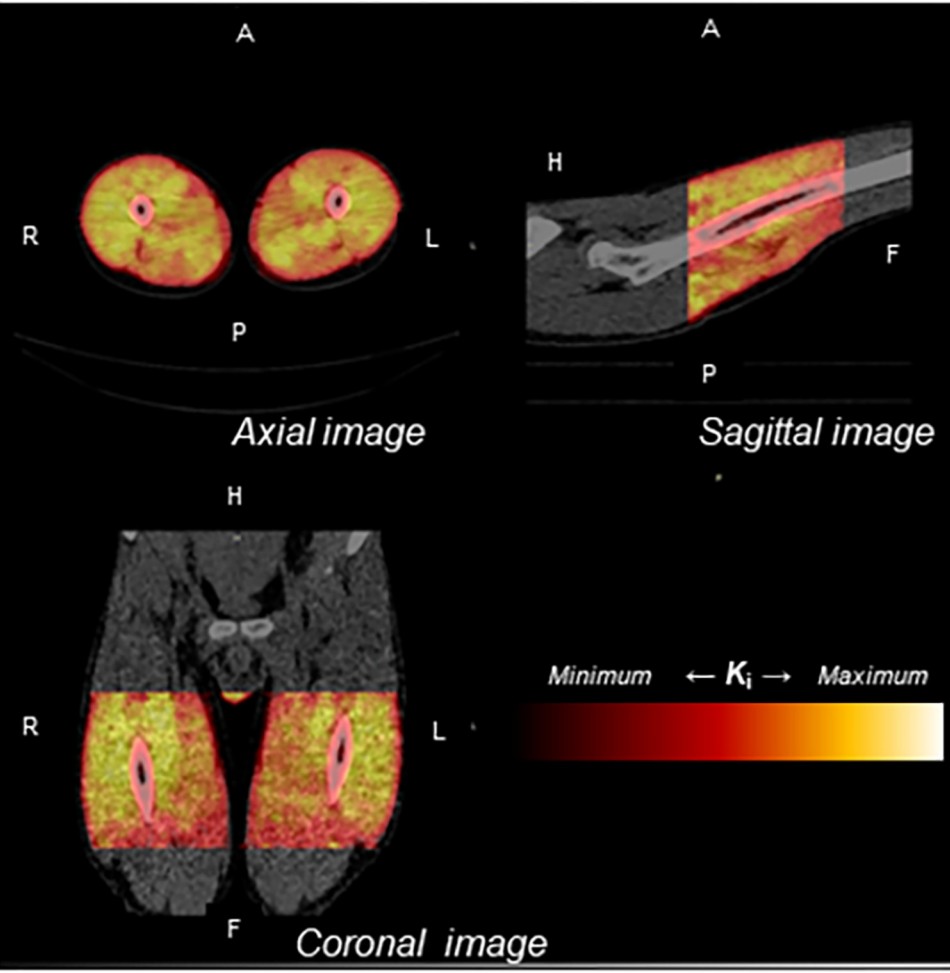

**Fig 6. Visualization of MPS by voxel-by-voxel *Ki* heatmapping.** Voxel-by-voxel heatmap of the $K_i$ overlaying the CT image is shown. $K_i$ was presented as a heatmap, in which the maximum values are white and the minimum values black. $K_i$ was relatively high in the quadriceps but low in the hamstrings.

higher $K_i$ after ingestion of yogurt containing 20 g milk protein than after ingestion of water (substantially fasting state) in 5 of the 6 participants suggests that our trial was successful. This non-invasive method has benefits especially in frail individuals, such as the elderly and patients with chronic disease. However, these results were unsatisfactory due to the small, non-significant difference in the mean $K_i$ between milk protein intake and fasting. A previous report evaluating 25 g whey protein showed a significant increase in $K_i$ calculated by similar methods to those in our study compared with fasting, and this effect size was estimated to be moderate [12]. Despite the clinical study by Arentson-Lantz et al. [12] who used whey protein, we used fermented whole milk protein because we hypothesized that fermented whole milk protein intake will moderately increase $K_i$ as seen in animal studies [21, 22].

The discrepancies between our hypothesis and the results of this investigation could be due to the following reasons. First, it is possible that $K_i$ was not calculated accurately due to body movements. In this study, the two PET/CT examinations were performed in the same supine position and resting condition, so the $K_i$ of the left and right thighs should be almost the same. It was estimated from the PET/CT images that the $K_i$ of 2 of the 6 participants might had been largely affected by body movements. In participants A and E, who had particularly large left–right differences, relatively large body movements were observed during dynamic PET imaging. Presumably, the significant correlations between each $K_i$ after intake of each test food were not affected by the outlier $K_i$ of the 2 participants with large body movements during PET/CT imaging. Fig 4C shows a PET/CT image of the right thigh of participants A. In the first test, high radioactive areas suggestive of the femoral artery were nearly constant throughout imaging, indicated by the intersecting blue lines at the upper right of each image in Fig 4. However, in the second test, the high radioactive area moved from the upper to lower right of the blue line intersection, and it was presumed that the femoral artery had moved out of the input VOI during dynamic PET imaging.

Second, differences in the type of protein ingested may have affected the results. Five of the 6 participants had a higher $K_i$ after milk protein ingestion than after fasting. It could be assumed that $K_i$ actually increased after protein ingestion, but the increase was masked by the effect of body movement. However, in the 3 participants whose $K_i$ was higher after protein intake than after water intake, the difference was less than 5%. Therefore, it is still possible that the MPS-enhancing effect was smaller after ingestion of fermented milk protein than whey protein.

Another consideration is the timing of the $K_i$ evaluation. In this study, dynamic PET imaging was performed using the same time schedule as that of a previous study [12], i.e., 90–150 min after ingestion of each test food. In the calculation of $K_i$ from dynamic PET images using Patlak graphical analysis, the first half period (90–120 min after ingestion of the test food) was the time required for equilibration between the free [$^{11}$C]Met content in the arterial blood and unlabeled Met in the muscle's free amino acid pool. The second half period (120–150 min after ingestion of the test food) involved the actual evaluation of the influx of [$^{11}$C]Met into muscle protein. In the elderly, it has been reported that the timing of the MPS increase is later than that of the peak blood amino acid level [23]. Therefore, assessment at 120–150 min after protein intake is reasonable. On the other hand, the timing of the increase in MPS is faster in young people than in the elderly, such that the peaks in the blood amino acid level and MPS almost coincide [24]. In a clinical study using the same yogurt as in this study, the concentrations of blood amino acids including leucine peaked at 45 min after ingestion, decreased to half the peak concentrations after 90 min, and decreased further to approximately one-quarter the peak concentrations after 120 min [25]. Therefore, the timing of imaging after protein intake should also be considered, taking into account age and the type of protein. The primary objective of this study was to investigate MPS in living humans; therefore, the selected time-

scale seemed appropriate and reasonable based on existing research and the physiological context of our investigation. However, further validation is required to confirm the time-scale in future studies.

When evaluating MPS with PET/CT, it may also be necessary to consider comparing [$^{11}$C]Met with other amino acid radiotracers, such as L-[$^{11}$C]leucine ([$^{11}$C]Leu). However, [$^{11}$C]Leu is metabolically active in skeletal muscle, which makes it complicated to estimate the protein synthesis rate. Previous studies have extensively used [$^{11}$C]Leu as a radiotracer to estimate protein synthesis rates in skeletal muscle before our initial site-specific MPS estimation. Therefore, it may be beneficial to validate the accuracy of MPS evaluation using [$^{11}$C]Met in the future by comparing it to the MPS evaluation using [$^{11}$C]Leu. In future research, further efforts to minimize body movements, improve the acquisition sites, introduce techniques to correct the effects of body movements after imaging, and optimize the timing of $K_i$ evaluation, are warranted.

Moreover, we noted another advantage of this method to evaluate all skeletal muscle regions, including deep and/or relative thin skeletal muscle where it is difficult to perform biopsy. In this study, when $K_i$ was calculated separately for two muscle regions, the MPS rate was higher in the quadriceps than in the hamstrings. Also, there was a high positive correlation in the $K_i$ between the quadriceps and hamstrings after intake of water or milk protein. These results suggest that MPS differed between the skeletal muscle regions, but these regional MPS differences among individuals represented the entire region. This seems reasonable because even in the same body part (e.g., the leg), MPS during the basal state and anabolic states, such as after exercise, differed for each skeletal muscle due to differences in muscle fiber composition and motor function activity in animal experiments [13–16]. In current clinical evaluations of MPS, the quadriceps femoris muscle, the largest superficial muscle group in the thigh, which regulates knee joint extension function, is often assessed. Furthermore, since muscle biopsies at multiple sites are required, studies of each muscle site are limited [26]. On the other hand, it has been shown that functional deterioration of the hamstrings, which was the focus of this study, leads to injuries such as muscle strain in athletes [27] and is related to pain and/or mobility in patients with osteoarthritis [28]. Therefore, there should be sufficiently high demand for low-burden methods, such as PET/CT, to evaluate interventions aimed at maintaining site-specific muscle mass and function, including in the hamstrings. The deep region anatomy of the hamstrings makes it difficult to conduct skeletal muscle biopsies. The present method is a low-invasive way to evaluate the entire thigh muscle region.

Recently, it was reported that sarcopenia occurs predominantly in the thigh, particularly the quadriceps region [29]. This suggests that age-related muscle atrophy may involve site-specific MPS fluctuations. Our results showed that MPS was higher in the quadriceps than hamstrings in young adults. Therefore, it would be interesting to examine the possibility that the effect of MPS attenuation with aging is particularly large in the quadriceps region using this method in the future. In the future, PET combined with a high-resolution reference imaging system such as magnetic resonance imaging (MRI) might be used to evaluate $K_i$ by skeletal muscle region [30, 31]. Therefore, it would be important to clarify the research aim or evaluation focus and then select the method, e.g., PET/CT, PET/MRI, or muscle biopsy, based on their advantages.

Furthermore, we attempted to visualize estimated MPS using an image modeling technology to express the $K_i$ intensity for each voxel as a heatmap. As a result, we found higher $K_i$ in the quadriceps than in the hamstrings. Thus, we demonstrated the possibility of visualizing whole-region MPS as a 3D image, although it will be necessary to verify the validity of expanding the Patlak plot model calculated from analyzed PET slices to the entire region in the future. Moreover, visualization of estimated MPS over the entire acquisition region, including outside

the researchers' area of interest, may provide unexpected insights into social issues such as sarcopenia. Given these aspects, our approach using [$^{11}$C]Met PET/CT to assess MPS in humans offers unique insights, despite the absence of a significant increase in $K_i$.

Our study has limitations. Because this was a pilot study with a small sample size, we should validate our findings in a larger sample size to improve the methods for a full-scale experiment. As a next step, it would be beneficial to conduct further studies with a larger sample size or under various circumstances to evaluate $K_i$ change more accurately.

In conclusion, this pilot study presents unique advantages of the image-derived MPS estimation method using dynamic PET after milk protein ingestion, focusing on site-specific evaluation and visualization of MPS. This non-invasive method might be beneficial for evaluating MPS in those suffering from sarcopenia such as the elderly and patients with chronic disease. This non-invasive method could contribute to overcoming the challenges of our aging society in the future.

## Acknowledgments

We express special thanks to all the clinical research assistants and laboratory assistants at QST and Meiji Co., and to M.R. Zhang, Ph.D. and K. Kawamura, Ph.D., who completed the timely production of [$^{11}$C]Met.

## Author Contributions

**Conceptualization:** Koichiro Sumi.

**Data curation:** Koichiro Sumi, Ryuichi Nishii.

**Formal analysis:** Koichiro Sumi, Kana Yamazaki, Ryuichi Nishii.

**Investigation:** Koichiro Sumi, Kana Yamazaki, Ryuichi Nishii, Misato Sakuda, Kentaro Tamura.

**Methodology:** Koichiro Sumi, Kana Yamazaki, Ryuichi Nishii, Misato Sakuda.

**Project administration:** Ryuichi Nishii, Kentaro Nakamura, Kinya Ashida, Tatsuya Higashi.

**Resources:** Kentaro Nakamura, Kinya Ashida.

**Supervision:** Tatsuya Higashi.

**Validation:** Koichiro Sumi.

**Visualization:** Koichiro Sumi.

**Writing – original draft:** Koichiro Sumi, Kana Yamazaki, Ryuichi Nishii.

**Writing – review & editing:** Koichiro Sumi, Kana Yamazaki, Ryuichi Nishii, Misato Sakuda, Kentaro Nakamura, Kinya Ashida, Kentaro Tamura, Tatsuya Higashi.

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
