## [Decision Letter · Decision Letter 0]

10 Mar 2024

PONE-D-23-43762Novel advantages of 11C-L-methionine dynamic PET/CT for assessing the rate of skeletal muscle protein synthesis: A pilot trial in young Japanese menPLOS ONE

Dear Dr. YAMAZAKI,

Thank you for submitting your manuscript to PLOS ONE. After careful consideration, we feel that it has merit but does not fully meet PLOS ONE’s publication criteria as it currently stands. Therefore, we invite you to submit a revised version of the manuscript that addresses the points raised during the review process.

We look forward to receiving your revised manuscript.

Kind regards,

Juan J Loor

Academic Editor

PLOS ONE

Journal Requirements:

2. Thank you for stating the following in the Competing Interests section: "I have read the journal's policy and the authors of this manuscript have the following competing interests: [K. S., M. S., K. N., and K. A. are salaried employees of Meiji Co., Ltd., a food company, which also provided experimental funding, materials, facility, and staff assistance to this research. But, Meiji Co., Ltd. did not play any role in the study design, data collection and analysis, decision to publish, or preparation of the manuscript. K. Y., R. N., K. T., and T. H. has no conflicts of interest to declare.]"

We note that you received funding from a commercial source: Meiji Co., Ltd

Within this Competing Interests Statement, please confirm that this does not alter your adherence to all PLOS ONE policies on sharing data and materials by including the following statement: ""This does not alter our adherence to PLOS ONE policies on sharing data and materials.” (as detailed online in our guide for authors http://journals.plos.org/plosone/s/competing-interests).  If there are restrictions on sharing of data and/or materials, please state these. Please note that we cannot proceed with consideration of your article until this information has been declared. 

3. In the online submission form, you indicated that "Raw data cannot be shared publicly because it is stored as an electronic medical record containing personal information in QST hospital. The anonymized data that support the findings of this study are available from the corresponding author, K. Y., upon reasonable request."

Reviewers' comments:

Reviewer's Responses to Questions

**Comments to the Author**

1. Is the manuscript technically sound, and do the data support the conclusions?

Reviewer #1: Partly

Reviewer #2: Yes

2. Has the statistical analysis been performed appropriately and rigorously? 

Reviewer #1: Yes

Reviewer #2: Yes

3. Have the authors made all data underlying the findings in their manuscript fully available?

Reviewer #1: Yes

Reviewer #2: Yes

4. Is the manuscript presented in an intelligible fashion and written in standard English?

Reviewer #1: Yes

Reviewer #2: Yes

5. Review Comments to the Author

Reviewer #1: The aim of the present manuscript is to evaluate the muscle protein synthesis (MPS) using dynamic 11C-methionine PET/CT in 6 healthy subjects. The study may be of interest to readers, even though the small sample size may reduce the reproducibility and the accuracy of these data.

Aside from a general improvement in the English language, the papers lacks a paragraph describing the limitations of the study, which I suggest including before considering it suitable for publication in PLOS.

Reviewer #2: Authors demonstrated dynamic PET/CT using 11C-radiolabeled L-methionine in order to assess the rate of skeletal muscle protein synthesis.

I really understand that authors’ PET studies are very important and unique from the viewpoint of medical trials for establishing non-invasive quantitative analysis of skeletal muscle protein synthesis in men. Authors’ research strategy using the radiotracer [11C]methionine can be apprecited as a suitable approach by medical scientist because methionine serves as the initiating amino acid in the process of protein synthesis.

However, from the general scientific view, I think that authors’ work is still in a preliminary stage and lack of scientific originality. Actually, authors did not obtain the result of significant increase in Ki after milk protein ingestion. There is less information about the quality control of the [11C]methionine injection solution such as purity, volume, and molar activity. Further, there are some concerns about the start-time of in vivo muscle protein synthesis and its time-scale of PET scan with [11C]methionine after milk protein ingestion.

Some other medical groups try to perform similar studies of some biological functions like in vivo protein synthesis using [11C]methionine which is a conventional radiotracer. If considered another possibility of the trial, parallel use of radioactive [11C]methionine and [11C]luecine might be encouraged for the assessment, the latter of which is an activator of the mammalian target of rapamycin in association with the protein synthesis.

From the comprehensive standpoint, I do not think that this article meets the journal quality and philosophy of PLOS ONE, which is a relative leading-edge journal. It might be better that this article would be summarized as a letter/communication style and submitted to more specific journals regarding PET clinical study.

P.S.

According to the IUPAC nomenclature, description of L-[11C]methionine is preferred to 11C-L-methionine.

6. PLOS authors have the option to publish the peer review history of their article (what does this mean?). If published, this will include your full peer review and any attached files.

Reviewer #1: No

Reviewer #2: No

---

## [Author Response · Author response to Decision Letter 0]

19 Apr 2024

We thank you for giving us the opportunity to make revisions. We sincerely appreciate it.

---

## [Decision Letter · Decision Letter 1]

3 Jun 2024

Unique advantages of dynamic L-[11C]methionine PET/CT for assessing the rate of skeletal muscle protein synthesis: a pilot trial in young men

PONE-D-23-43762R1

Dear Dr. YAMAZAKI,

We’re pleased to inform you that your manuscript has been judged scientifically suitable for publication and will be formally accepted for publication once it meets all outstanding technical requirements.

Kind regards,

Juan J Loor

Academic Editor

PLOS ONE

Additional Editor Comments (optional):

Reviewers' comments:

Reviewer's Responses to Questions

**Comments to the Author**

1. If the authors have adequately addressed your comments raised in a previous round of review and you feel that this manuscript is now acceptable for publication, you may indicate that here to bypass the “Comments to the Author” section, enter your conflict of interest statement in the “Confidential to Editor” section, and submit your "Accept" recommendation.

Reviewer #2: (No Response)

2. Is the manuscript technically sound, and do the data support the conclusions?

Reviewer #2: Yes

3. Has the statistical analysis been performed appropriately and rigorously? 

Reviewer #2: Yes

4. Have the authors made all data underlying the findings in their manuscript fully available?

Reviewer #2: Yes

5. Is the manuscript presented in an intelligible fashion and written in standard English?

Reviewer #2: Yes

6. Review Comments to the Author

Reviewer #2: The revised manuscript is well-written.

I think that the revised manuscript meets the quality and criteria of PLOS ONE.

7. PLOS authors have the option to publish the peer review history of their article (what does this mean?). If published, this will include your full peer review and any attached files.

Reviewer #2: No

---

## [Editor Report · Acceptance letter]

21 Jun 2024

PONE-D-23-43762R1 

PLOS ONE

Dear Dr. Yamazaki, 

I'm pleased to inform you that your manuscript has been deemed suitable for publication in PLOS ONE. Congratulations! Your manuscript is now being handed over to our production team.

Kind regards, 

on behalf of

Dr. Juan J Loor 

Academic Editor

PLOS ONE